# Examining Cultural Differences in the Associations between Appraisals and Emotion Regulation and PostTraumatic Stress Disorder in Malaysian and Australian Trauma Survivors

**DOI:** 10.3390/ijerph19031163

**Published:** 2022-01-21

**Authors:** Laura Jobson, Shamsul Haque, Siti Zainab Abdullah, Bryan Lee, James Haoxiang Li, Tamsyn Reyneke, Britney Kerr Wen Tan, Winnie Lau, Belinda Liddell

**Affiliations:** 1Turner Institute for Brain and Mental Health and School of Psychological Sciences, Monash University, Clayton, VIC 3800, Australia; jjlee32@student.monash.edu (B.L.); hlii0059@student.monash.edu (J.H.L.); trey0001@student.monash.edu (T.R.); 2Department of Psychology, Jeffrey Cheah School of Medicine and Health Sciences, Monash University Malaysia, Subang Jaya 47500, Malaysia; shamsul@monash.edu (S.H.); sabd0013@student.monash.edu (S.Z.A.); btan0062@student.monash.edu (B.K.W.T.); 3Phoenix Australia-Centre for Posttraumatic Mental Health, Department of Psychiatry, University of Melbourne, Carlton, VIC 3053, Australia; wlau@unimelb.edu.au; 4School of Psychology, University of New South Wales, Sydney, NSW 2052, Australia; b.liddell@unsw.edu.au

**Keywords:** culture, trauma, post-traumatic stress disorder, appraisals, emotion regulation, self-construal

## Abstract

Appraisals and emotional regulation play a central role in posttraumatic stress disorder (PTSD). Despite research demonstrating cultural differences in everyday appraisals and emotion regulation, little research has investigated the influence of culture on these processes in PTSD. This study examined cultural differences in the associations between appraisals, emotion regulation and PTSD symptoms using trauma survivors from an individualistic Western culture (Australia) and a collectivistic Asian culture (Malaysia). Trauma survivors (*N =* 228; 107 Australian with European cultural heritage, 121 Malaysian with Malay, Indian or Chinese cultural heritage) completed an on-line survey assessing PTSD (PTSD Checklist for the DSM-5 with Life Events Checklist), appraisals (trauma-related, fatalism, cultural beliefs about adversity) and emotion regulation (suppression, reappraisal, interpersonal). The Malaysian group reported significantly greater fatalism, cultural beliefs about adversity, suppression and interpersonal emotion regulation than the Australian group. Greater trauma-specific appraisals, greater suppression, fewer cultural beliefs about adversity, and less use of social skills to enhance positivity were generally associated with greater PTSD symptom severity, with little evidence of cultural group moderating these associations. Interdependent self-construal mediated the relationships between cultural adversity beliefs, enhanced positivity, reappraisal, perspective taking and PTSD symptoms. Independent self-construal mediated the relationships between fatalism and perspective taking and PTSD symptoms. Cultural group did not moderate these indirect effects. Interdependent self-construal mediated the associations between interpersonal regulation strategies of soothing and social modelling with PTSD symptoms for the Malaysian but not the Australian group. These findings demonstrate the importance of considering self-construal and culture in understanding factors associated with PTSD.

## 1. Introduction

Posttraumatic stress disorder (PTSD) is a significantly disabling psychiatric condition observed in most societies and cultures [1]. Trauma-related appraisals and emotion regulation difficulties contribute to the development and maintenance of PTSD [2,3,4]. Given appraisals and emotion regulation are both identifiable, and potentially modifiable, these two processes are important targets in current evidence-based psychological treatments for PTSD [1,2,5]. Decades of research has established the types of appraisals and emotion regulation strategies associated with PTSD in those from Western cultural backgrounds. This existing evidence-base informs our current PTSD psychological treatments. However, this evidence is derived predominately from Western, English-speaking patients, and largely ignores trauma survivors from other cultural groups [6]. Hence, most existing PTSD treatments are based on Western cultural norms, beliefs and values [6,7,8]. Consequently, culturally tailored PTSD interventions are scarce [6,7,8]. Given the debilitating nature of PTSD on the lives of individuals and their families, there is an urgent need to develop culturally informed PTSD treatments, especially as treatment effects improve significantly when culturally tailored [9].

Cognitive appraisals are one of the most useful predictors of PTSD [3]. Negative beliefs about the self (e.g., “I will never be the same again”), self-blame (e.g., “The event happened because of the way I acted”) and negative beliefs about the world (e.g., “The world is against me”) are all strongly associated with PTSD [2,10]. A meta-analysis identified strong associations between PTSD and problematic emotion regulation strategies, such as suppression of emotion and poor reappraisal [4]. Unfortunately, the majority of PTSD appraisal research and all studies included in the emotion regulation meta-analysis focused on trauma survivors from Western cultural backgrounds [4,7]. This is a concern because non-PTSD studies have shown that different cultures substantially differ in how appraisals are constructed and which emotion regulation strategies are effective in reducing negative affect [11,12].

Culture influences appraisals and emotion regulation through social orientations, beliefs, and values [11]. Substantial non-clinical research has focused on significant differences between those from Western and Asian cultural backgrounds. A key cultural variable, self-construal, is proposed to influence how members of Western and Asian cultures differently appraise experiences and regulate emotion. Western cultures tend to conceptualise the self as independent, autonomous, self-expressive, and promoting personal goals [13]. In contrast, in Asian cultures, these individualised aspects are less relevant to one’s self-concept. Instead, there is a greater emphasis on relatedness, interdependence, and the way that the self attends to and fits in with others and the surrounding social context [13]. 

Cultural differences in self-construal lead to differences in how individuals interpret life experiences [7]. Those from Western cultures tend to appraise events through the lens of their personal control, with a focus on agency and personal accomplishment. By contrast, personal agency and control have limited applicability in Asian cultures [7,13,14]. Rather fatalism (i.e., acceptance of the situation and the belief that destinies are ruled by an unseen power or inevitably played out) [15] and specific cultural beliefs about adversity (i.e., beliefs which emphasise the positive value of adversity, people’s capacity to overcome adversity and people’s inability to change adversity and its negative impacts) [16] are proposed to be of greater relevance in Asian cultures e.g., [7,16] and may be associated with greater PTSD symptoms [15]. Despite this, such appraisals have rarely been considered in the context of PTSD. Appraising events in terms of the personal self, control and achievement is important for the general wellbeing of people from a Western background but has less relevance to the wellbeing of those from Asian backgrounds [7,17]. Given the focus of appraisals in PTSD treatments, these cultural differences raise important questions for post-traumatic adjustment and recovery. 

Culture shapes individual preferences for emotional states and the strategies used to regulate emotions. Those from Asian cultures tend to value moderation, suppression and control of intense emotional experiences [11,12]. However, in Western cultures, there is often an emphasis on the experience, expression and making sense of emotions [11,12]. These differences lead to cultural variability in which emotion regulation strategies are deemed adaptive or maladaptive [11,12]. Asian cultures promote emotion regulation approaches that encourage disengaging from strong emotions (e.g., suppression of emotion), whereas those from Western cultures tend to focus on emotion engagement strategies (e.g., expression and making sense of emotion) [12,14]. Thus, in Asian cultures suppression is not necessarily associated with poor mental health but engaging in suppression in Western cultures appears to promote negative emotions [12]. Additionally, most emotion regulation research has focused exclusively on *intrapersonal* processes (which tends to be more focused on the independent aspect of self) and much less attention has been given to examining *interpersonal* emotion regulation processes (which tends to be more focused on the interdependent aspect of self) [18]. Recently, researchers have commenced investigating interpersonal emotion regulation, which utilizes social cues to facilitate emotion regulation (enhancing positive affect, perspective taking, soothing, and social modelling) [18].

There has been some research investigating appraisals and emotion regulation in culturally distinct groups. Western trauma survivors with PTSD have been found to have fewer appraisals of control and greater mental defeat and sense of permanent negative self-change than those without PTSD, while these appraisals were not associated with PTSD in those from Asian cultural backgrounds [19,20]. Additionally, Bernardi and Jobson found that culture moderated the relationship between PTSD symptoms and appraisals of mental defeat, personal control, self-blame, and mastery, such that these correlations were significantly stronger among European Australian trauma survivors than for Asian Australian trauma survivors [21]. Additionally, Bernardi and Jobson found that independent self-construal specifically mediated these associations and cultural group affected the strength of these relationships [21]. 

European Australian trauma survivors with PTSD have been found to report less use of individualistic reappraisal strategies, and greater use of suppression, worry, and emotion dysregulation than European Australian trauma survivors without PTSD. For Asian Australian trauma survivors these regulation strategies were not associated with PTSD symptoms [22]. It has recently been proposed that Asian cultures that value interdependence may find interpersonal emotion regulation strategies to be particularly effective in reducing negative affect [23]. In support of this, Liddell and Williams found that East Asian students reported higher use of interpersonal emotion regulation strategies in daily life as compared to Western European students and engaging in interpersonal emotion regulation when exposed to a stressor was more beneficial for East Asian than Western participants [23]. However, these studies have all been conducted using Asian samples living in Western cultures, which raises questions about acculturation and generalizability to Asian cultures. Moreover, much of this research has demonstrated that Western-focused appraisals and emotion regulation strategies have less relevance for trauma survivors from Asian cultural backgrounds, e.g., [19,20,21,22]. Thus, important questions still remain regarding what appraisals and emotion regulation strategies *are of relevance* for trauma survivors with PTSD from Asian cultural backgrounds. 

### Current Study

The aim of this study was to further investigate cultural differences in the associations between appraisals, emotion regulation and PTSD using a cross-country design. We selected trauma survivors from an Asian, interdependent cultural group (Malaysia) and from a Western, independent cultural group (Australia) [24,25,26]. 

Malaysia is a Southeast Asian country. After more than a century of British colonial rule, Malaysia gained independence in 1957 [27]. Contemporary Malaysian society is influenced by Malay, Chinese and Indian values and traditions [28]. Around half of the population of Malaysia is Malay, with minorities of Indian, Chinese and indigenous peoples [29]. Malaysia’s official language is Malay and official religion is Islam [28,29]. Malaysia tends to be accepting of hierarchical order in society and values social harmony and the honouring of cultural norms and traditions [26]. Of relevance to the current study, Malaysians tend to hold a fatalistic worldview [28,29], Malaysian culture has been significantly influenced by Chinese beliefs and values [30,31], and Malaysia is a collectivistic, interdependent society [26]. 

Australia was colonized by the British in the late 18th century, resulting in Aboriginal and Torres Strait Islander peoples being dispossessed of their lands and subject to genocidal practices and policies [32]. The social composition of the country was dramatically altered and a Western, European culture became dominant [33]. The first half of the 20th century focused on facilitating ‘white’ migration [34]. However, in modern Australia migrants come from Asia, the Americas and Africa, with approximately 26% of Australian residents being born overseas [33]. The official language of Australia is English and Australia is considered a secular country, with Christianity being the dominant religion introduced by the British colonial settlers [33]. Australian society values emphasise egalitarianism [26]. Australia is a highly individualistic society [26]. 

We aimed to examine appraisal types (fatalism, specific cultural beliefs about adversity) and emotion regulation strategies (suppression, interpersonal emotion regulation strategies) proposed to be associated with interdependence and valued by those from Asian cultural backgrounds [7,23], but rarely investigated in the PTSD literature. We have also included several appraisal and emotion regulation types (negative self, self-blame, reappraisal) that relate to independence and thus should be more highly endorsed by Australian trauma survivors. 

We hypothesized that the Malaysian group would report greater fatalism, cultural beliefs about adversity, suppression and interpersonal emotion regulation and less negative self, self-blame, and reappraisal than the Australian group (Hypothesis 1). Second, we predicted that these appraisals and emotion regulation types would be associated with PTSD symptoms, given their identified role in the development and maintenance of PTSD (Hypothesis 2). However, given cultural differences in the value of these strategies, we predicted culture would moderate these associations (Hypothesis 3). Specifically, the associations between negative independent appraisals and maladaptive emotion regulation strategies (i.e., negative self, self-blame, poorer reappraisal, greater suppression) and PTSD symptoms would be stronger for the Australian group than the Malaysian group. In contrast, the associations between negative interdependent appraisals and maladaptive emotion regulation strategies (i.e., fatalism, cultural beliefs about adversity, poorer interpersonal emotion regulation, lower suppression) and PTSD symptoms would be stronger for the Malaysian group than the Australian group. 

Finally, following the approach of Bernardi and Jobson [21], we investigated moderating mediation models in which appraisals/emotion regulation were the independent variable, PTSD symptoms the dependent variable, self-construal (independent and interdependent) would function as mediators, and group (i.e., Malaysian, Australian) would act as a moderator of the indirect effects (see Figure 1). We predicted that independent self-construal would mediate the relationship between independent self-construal variables and PTSD. Interdependent self-construal would mediate the relationship between interdependent self-construal variables and PTSD. We predicted cultural group would moderate these conditional effects; the interdependent indirect effects would be stronger for the Malaysian group than the Australian group and the independent indirect effects would be stronger for the Australian group than the Malaysian group.

## 2. Materials and Methods

### 2.1. Design

The study obtained ethical approval from the Monash University Human Research Ethics Committee (2021-27577-58747). The study employed a cross-country, cross-sectional design. Australian and Malaysian researchers co-designed the study to ensure cultural appropriateness of the design, data analysis and interpretation, and dissemination of findings.

### 2.2. Participants

Participants were recruited from the general community in Australia and Malaysia using social media adverts (Facebook, Gumtree). The study inclusion criteria were: (a) having experienced a criterion A trauma experience (as indexed by the Life Events Checklist [35], (b) Australian participants identifying as having European heritage (i.e., all four grandparents of European heritage) and Malaysian participants identifying as having Malay, Chinese or Indian heritage (i.e., all four grandparents of Malay, Chinese or Indian heritage), (b) being aged between 18 and 65 years, and (c) able to complete the online survey in either English or Malay. Exclusion criteria included rapid responders (i.e., those who completed the survey in under 10 min), scoring below the conscientious response cut-off, and completing the survey more than once.

Of the 141 Australian responders, 34 were excluded (no trauma exposure: *n* = 2; rapid responders: *n* = 26; failed to meet the conscientious responses cut-off: *n* = 4; completed the survey twice: *n* = 2). Of the 170 Malaysian responders, 49 were excluded (no trauma exposure: *n* = 24; rapid responders: *n*= 3; failed to meet the conscientious responses cut-off: *n* = 21; completed the survey twice: *n* = 1). The final sample consisted of 228 (107 Australian and 121 Malaysian) trauma survivors. 

A G*Power analysis was used to estimate sample size. These estimates were based on the moderation analyses using small to moderate effect sizes [21,22], alpha of 0.05, and 80% power. It was estimated that the study required 101 participants per group. Given the novelty of the moderated mediation analyses, it was difficult to ascertain sample size estimates for these analyses and thus, these analyses were exploratory. 

### 2.3. Measures

#### 2.3.1. Trauma Exposure and PTSD Symptoms

**Life Events Checklist (LEC)** [35]. The LEC contains 17 self-report items that screen for life-time exposure to potentially traumatic events using 6-point nominal scales (e.g., “happened to me”, “learned about it”, “part of my job”) [35]. In the current study it provided a screener for trauma exposure and contextual information (trauma type and time of trauma) for the index trauma. 

**PTSD Checklist for the DSM-5 with Life Events Checklist (PCL-5)** [35]. The PCL-5 is a 20-item self-report measure of PTSD symptoms in response to the index trauma reported on the LEC. Items are scored on 5-point Likert scales (0 = *not at all* to 5 = *extremely*) and responses are summed to provide a total PTSD severity score (total score range: 0–80), with higher scores indicating greater PTSD symptom severity [35]. A PCL-5 cut-point score of 33 has been suggested as a reasonable value to use for a provisional PTSD diagnosis [35]. The PCL-5 has good psychometric properties, including discriminant and convergent validity, and test–retest reliability [36], including in cross-cultural research [37]. In the present study, the PCL-5 yielded excellent internal consistency for the Australian group (α= 0.95) and Malaysian group (α= 0.96). 

#### 2.3.2. Appraisals

**PostTraumatic Cognitions Inventory (PTCI)** [10]. The PTCI is a 33-item measure that assesses trauma-related appraisals. The PTCI includes three subscales; appraisals about negative self, negative world and perceived self-blame regarding the trauma. Items are rated on 7-point scales (1 = *totally disagree* to 7 = *totally agree*). The PTCI is a well-established inventory [10] and has been used in cross-cultural research [21,38]. In the current study the total scale and subscales demonstrated good internal consistency (Australia PTCI total α= 0.97, negative self α= 0.95, negative world α= 0.93, self-blame α= 0.91; Malaysia PTCI total α = 0.97, negative self α = 0.96, negative world α = 0.90, self-blame α = 0.79).

**Fatalism Questionnaire** [15,39]. The Fatalism Questionnaire includes six items that assess an individual’s propensity to believe that one’s destiny is externally determined. Response options range from 1 (*strongly disagree*) to 5 (*strongly agree*). The measure has good test–retest reliability (*r* = 0.71) and external validity [39]. The questionnaire has been used cross-culturally [15,39]. In the current study, internal consistency was good (Australia α = 0.76; Malaysian α = 0.84).

**Chinese Cultural Beliefs about Adversity Scale (CBA)** [16]. The CBA assesses specific cultural beliefs about adversity. Malaysia has a large Chinese population and Malaysian culture has been significantly influenced by Chinese beliefs and values [30,31]. Thus, it was included as a cultural measure of beliefs about adversity. It contains nine items, with two items focused on negative cultural beliefs about adversity (which are reversed scored) and the remaining items focused on positive cultural beliefs about adversity. For each item respondents indicate the degree to which they agree with the item on a 6-point Likert scale. Higher scores indicate a higher degree of agreement with positive cultural beliefs about adversity. Reliability for this measure has been found to be adequate (α = 0.76) [16]. In the current study, internal consistency was adequate (Australia α = 0.65; Malaysian α = 0.75).

#### 2.3.3. Emotion Regulation

**Interpersonal Emotion Regulation Questionnaire (IERQ)** [18]. The IERQ contains four subscales (5 items each); enhancing positive affect (i.e., which describes a tendency to seek out others to increase feelings of happiness and joy), perspective taking (i.e., which involves the use of others to be reminded not to worry and that others have it worse), soothing (i.e., which consists of seeking out others for comfort and sympathy) and social modelling (i.e., involves looking to others to see how they might cope with a given situation). The questionnaire has been found to have good psychometric properties (α’s between 0.89 and 0.94) [18]. In the current study, internal consistency was good (Australia enhance positivity α = 0.80, perspective taking α = 0.87, soothing α = 0.88, social modelling α = 0.80, total IERQ α = 0.90; Malaysian enhance positivity α = 0.89, perspective taking α = 0.83, soothing α = 0.90, social modelling α = 0.83, total IERQ α = 0.95).

**Emotion Regulation Questionnaire (ERQ)** [40]. The ERQ is a 10-item scale that measures respondents’ tendency to regulate their emotions in two ways: (1) cognitive reappraisal, and (2) expressive suppression. Respondents answer each item on 7-point Likert-type scales (1 = *strongly disagree* to 7 = *strongly agree*). The ERQ is a routinely used measure emotion regulation, including in cross-cultural research [22], and has good psychometric properties [40]. In the current study, internal consistency was good (Australia reappraisal α = 0.83, suppression α = 0.75; Malaysian reappraisal α = 0.87, suppression α = 0.77).

#### 2.3.4. Other Measures

**Self-Construal Scale (SCS)** [41,42]. The SCS is a 30-item scale that measures how people view themselves in relation to others. It is comprised of two sub-scales; independent self-construal (15 items) and interdependent self-construal (15 items). Participants respond to 30 statements about themselves on 7-point rating scales (1 = *strongly disagree* to 7 = *strongly agree*). Scores are totalled for each respondent providing an independent subscale score and an interdependent subscale score. This scale is widely used in cross-cultural research [41,42]. In the current study, internal consistency was good (Australia independent α = 0.79, interdependent α = 0.79; Malaysian independent α = 0.83, interdependent α = 0.77).

**Hospital Anxiety and Depression Scale (HADS)** [43]. The HADS was used to assess symptoms of anxiety and depression. The HADS consists of 14 items; 7 items assess depression symptoms and 7 items assess anxiety. Each item is scored on 4-point Likert scales assessing symptoms over the past week. Item responses are summed (some items are reverse scored) and range from 0 to 21, with higher scores indicating greater depression and anxiety symptom severity. Studies suggest a cut-off score of ≥8 for both anxiety and depression to be optimal for specificity and sensitivity [44]. Previous literature has found the HADS to have good internal consistency [45] and to be a psychometrically valid measure of depression and anxiety in many cultures and languages [45]. In the present study, the HADS yielded good reliability (Australia α = 0.72; Malaysia α = 0.77). 

**Conscientious Responder Scale (CRS) [46]**. The CRS is a validity scale that comprises five self-report items that are inserted randomly throughout a questionnaire. The CRS is used to differentiate between conscientious and indiscriminate response on a survey. Each item instructs participants how to respond (sample item: “Please answer this question by choosing option number two, “disagree.””). Items are scored on a 5-point Likert scale (1 = *strongly disagree* to 5 = *strongly disagree*). Correct responses are scored as “1”, while incorrect responses are scored as “0”. Item scores are summed, with higher scores indicating more conscientious responses and lower scores indicating indiscriminate responses. Previous literature and binomial probability have classified responses of 0–2 as indicative of indiscriminate responses [46]. Thus, the cut-off score for conscientious responders in the present study was a minimum of 3 correct responses.

### 2.4. Procedure

Individuals interested in finding out more about the study and/or participating contacted the researchers. The researchers provided these individuals with a link to the online survey, which was hosted on Qualtrics. At the commencement of the survey participants were provided with an explanatory statement outlining the details of the research. Participants provided informed consent by commencing the survey. Participants completed the LEC, PCL-5, HADS, PTCI, IERQ, ERQ, Fatalism Questionnaire, CBA, SCS and demographics (age, gender, education, ethnicity, and religion).

### 2.5. Data Analysis 

Prior to hypothesis testing, data cleaning was conducted using Microsoft Excel. All subsequent analyses were conducted using IBM SPSS Statistics 27. Missing data from both Australian and Malaysian groups was less than 5% and was replaced using the multiple imputation method with five imputed datasets and 10 iterations [47]. Z-transformations (z-score of ±3.29) identified that there were no outliers and Cook’s distance of more than 1 identified no multivariate outliers and independence of residuals was assumed (Durbin-Watson statistic was between 1 and 3) [47]. Assumptions of normality and homogeneity of variance were tested. Several variables were not normally distributed for both Australian and Malaysian groups and transformations did not improve normality. Therefore, bootstrapping with 5000 bootstrapped samples were used for all analyses [47]. All variables met the assumptions of linearity, homoscedasticity and homogeneity of variance. 

To assess Hypothesis 1, group differences were explored using two one-way (Malaysian vs. Australian) Multivariate Analysis of Covariances (MANCOVA), with the appraisal and emotion regulation strategies as the dependent variables. Due to group differences in age, education, religion and time since trauma (see below), these variables were included as covariates in all analyses that included cultural group comparisons. To examine Hypothesis 2, separate partial correlation analyses were used to assess the strength of the associations between appraisals and emotion regulation strategies and PTSD symptoms for the sample overall. Time since trauma was included as a covariate, given its influence on appraisals, emotion regulation and PTSD symptoms. We did not include age, education and religion as covariates in these correlation analyses, as these analyses were conducted for the whole sample (rather than for the separate cultural groups). However, it is worth noting that when these variables were included as covariates a similar pattern of results emerged. For the correlation analyses for the two separate groups, which are presented in Appendix A, time since trauma, age, religion and education were included as covariates. To control for multiple correlation analyses we used a Bonferroni corrected alpha of 0.005. 

To investigate Hypothesis 3, a series of separate moderated regression analyses were conducted using PROCESS (model 1) [48] with 5000 bootstrapped samples. Confidence intervals were used to determine significance of results, with confidence intervals not including 0 being considered significant. For our exploratory analyses, a series of moderated mediation models (see Figure 1) were tested in single models using bootstrapping to assess the significance of the indirect effects (self-construal) at differing levels of the moderator (cultural group) [48]. Appraisals/emotion regulation were the predictor variables, with self-construal as the mediators. The “PROCESS” macro (model 7) [48] with bias-corrected 95% confidence intervals (*n* = 5000) was used. The outcome variable was PTSD symptoms and cultural group was the proposed moderator. Moderated mediation analyses test the conditional indirect effect of a moderating variable (cultural group) on the relationship between a predictor (appraisals/emotion regulation) and an outcome variable (PTSD symptoms) via potential mediators (self-construal). Confidence intervals were used to determine significance of results, with confidence intervals not including 0 being considered significant.

## 3. Results

As shown in Table 1, the two cultural groups differed significantly in terms of age, education level, religion and time since trauma. Therefore, these variables were included as covariates in group comparisons. As predicted, the Malaysian group reported higher interdependent self-construal than the Australian group. However, contrary to predictions, the Malaysian group reported higher independent self-construal than the Australian group. Concerning provisional PTSD diagnosis, 28.97% of the Australian sample (*n* = 31) and 28.93% of the Malaysian sample (*n* = 35) scored above the clinical cut-off on the PCL-5.

### 3.1. Hypothesis 1: Cultural Group Differences

The MANCOVA assessing group differences in appraisal types revealed that the two groups differed significantly, Wilks’ Lambda = 0.96, *F*(5, 175) = 4.25, *p* = *0*.001, *η_p_*^2^ = 0.11. As shown in Table 2, in support of Hypothesis 1, follow-up analyses revealed the Malaysian group scored significantly higher than the Australian group in terms of fatalism appraisals and cultural beliefs about adversity. Contrary to Hypothesis 1, the groups did not differ significantly on trauma-related cognitions (as indexed by the PTCI). 

The MANCOVA assessing group differences in emotion regulation strategies revealed that the two cultural groups differed significantly, Wilks’ Lambda = 0.81, *F*(6, 201) = 7.76, *p* < 0.001, η_p_^2^ = 0.19. As shown in Table 2, in support of Hypothesis 1, follow-up analyses revealed the Malaysian group scored significantly higher than the Australian group in terms of suppression and interpersonal emotion regulation (perspective taking, social modelling and overall interpersonal emotion regulation). Contrary to Hypothesis 1, the Malaysian group reported significantly greater use of reappraisal strategies than the Australian group.

### 3.2. Hypothesis 2: Associations with PTSD Symptoms

When examining the associations between PTSD symptoms and appraisal types/emotion regulation strategies, we found that in line with Hypothesis 2, PTSD symptoms were significantly positively associated with self-blame, *r*(224) = 0.41, *p* < 0.001, 95%CI [0.29–0.53], negative self, *r*(224) = 0.65, *p* < 0.001, 95%CI [0.53–0.74], negative world, *r*(224) = 0.55, *p* < 0.001, 95%CI [0.44–0.63], and suppression, *r*(224) = 0.32, *p* < 0.001, 95%CI [0.18–0.44], and negatively associated with cultural beliefs about adversity, *r*(224)= −0.21, *p* = 0.002, 95%CI [−0.33–0.07], and using social cues to enhance positivity, *r*(224)= −0.21, *p* = 0.002, 95%CI [−0.34–0.06]. 

However, PTSD symptoms were not significantly associated with fatalism, *r*(224)= 0.09, *p* = 0.16, 95%CI [−0.04–0.23], reappraisal, *r*(224)= −0.06, *p* = 0.35, 95%CI [−0.21–0.07], perspective taking, *r*(224)= −0.02, *p* = 0.72, 95%CI [−0.16–0.11], soothing, *r*(224)= −0.08, *p* = 0.24, 95%CI [−0.22–0.06], or social modelling, *r*(224)= −0.09, *p* = 0.16, 95%CI [−0.23–0.05]. The correlation analyses for the two separate cultural groups are presented in Appendix A. The moderation analyses below examined cultural group differences in these presented associations.

### 3.3. Hypothesis 3: Moderation Analyses

When using moderation analyses to examine whether cultural group influenced the strength of the above associations, we found that there was little support for Hypothesis 3 (Here, we only present the results of the interactions [cultural group × appraisal/emotion regulation] in predicting PTSD symptoms. See Appendix A for full summary of moderation findings). Cultural group only moderated the association between self-blame and PTSD symptoms, *R*^2^ change = 0.04, *F*(1, 203)= 10.79, *p* = 0.001. Follow-up analyses revealed that the association was significantly stronger for the Malaysian group, effect = 1.47, *SE* = 0.22, *t* = 6.53, *p* < 0.001, 95%CI [1.03–1.91], than the Australian group, effect = 0.49, *SE* = 0.20, *t* = 2.51, *p* = 0.01, 95%CI [0.11–0.88] (see Figure 2).

Cultural group did not significantly moderate any of the other associations. The associations between PTSD and negative self, *R*^2^ change = 0.003, *F*(1, 203) = 0.98, *p* = 0.32, negative world, *R*^2^ change = 0.003, *F*(1, 203) = 0.99, *p* = 0.32, fatalism, *R*^2^ change = 0.005, *F*(1, 203) = 0.11, *p* = 0.75, cultural beliefs about adversity, *R*^2^ change < 0.001, *F*(1, 203) = 0.008, *p* = 0.93, reappraisal, R^2^ change = 0.01, *F*(1, 203) = 2.14, *p* = 0.15, suppression, *R*^2^ change = 0.004, *F*(1, 203) = 0.99, *p* = 0.32, enhance positivity, *R*^2^ change = 0.001, *F*(1, 203) = 0.31, *p* = 0.58, perspective taking, *R*^2^ change = 0.003, *F*(1, 203) = 0.70, *p* = 0.40, soothing, *R*^2^ change = 0.001, *F*(1, 203) = 0.20, *p* = 0.65, and social modelling, *R*^2^ change = 0.001, *F*(1, 203) = 0.15, *p* = 0.70, were all non-significant.

### 3.4. Exploratory Analyses—Moderated Mediation Analyses

In this section, we examined whether self-construal (independence and interdependence) mediated the relationships between appraisals/emotion regulation and PTSD symptoms, and whether cultural group moderated these indirect associations. Here, we present a summary of the significant findings (We only present the significant indirect pathway effects. See Appendix A for full outline of findings). Interdependent self-construal mediated the associations between cultural beliefs about adversity appraisals and PTSD symptoms, ß = 0.14, *SE* = 0.06, 95% CI [0.04, 0.28], reappraisal and PTSD symptoms, ß = 0.14, *SE* = 0.07, 95% CI [0.02, 0.28], enhanced positivity and PTSD symptoms, ß = 0.23, *SE* = 0.08, 95% CI [0.07, 0.39], and perspective taking and PTSD symptoms, ß = 0.19, *SE* = 0.09, 95% CI [0.01, 0.38]. However, there was no evidence of cultural group moderating these indirect effects.

Independent self-construal only mediated the relationships between fatalism appraisals and PTSD symptoms, ß = −0.12, *SE* = 0.07, 95% CI [−0.28, −0.01], and perspective taking and PTSD symptoms, ß = −0.15, *SE* = 0.08, 95% CI [−0.32, −0.01]. It is worth noting that these indirect effects were negative, indicating higher fatalism and perspective taking were associated with higher levels of independence which in turn were associated with lower levels of PTSD symptoms. There was also no evidence of cultural group moderating these conditional indirect effects.

Cultural group did moderate the conditional indirect effects for soothing, index = 0.24, *SE* = 0.13, 95% CI [0.02, 0.52], and social modelling, index = 0.22, *SE* = 0.13, 95% CI [0.01, 0.51]. Specifically, for the Malaysian group, interdependent self-construal mediated the relationships between soothing and PTSD symptoms, ß = 0.18, *SE* = 0.09, 95% CI [0.01, 0.38], and between social modelling and PTSD symptoms, ß = 0.24, *SE* = 0.16, 95% CI [0.04, 0.49]. However, these indirect effects were not observed for the Australian group; soothing ß = −0.06, *SE* = 0.06, 95% CI [−0.21, 0.05], social modelling, ß = 0.02, *SE* = 0.09, 95% CI [−0.13, 0.23].

## 4. Discussion

This study examined cultural differences in appraisals and emotion regulation strategies and their associations with PTSD symptoms. First, as expected, the Malaysian group reported significantly greater fatalism, cultural beliefs about adversity, suppression and interpersonal emotion regulation than the Australian group. Unexpectedly, the Malaysian group reported significantly greater reappraisal than the Australian group and the two groups did not differ significantly on trauma-specific appraisals. Second, as predicted, PTSD symptoms were significantly positively associated with self-blame, negative self, negative world, and suppression and negatively associated with cultural beliefs about adversity and using social cues to enhance positivity. However, inconsistent with predictions, cultural group only moderated the association between PTSD symptoms and self-blame. Third, generally, interdependent self-construal mediated the associations between cultural beliefs about adversity appraisals, reappraisal, enhanced positivity, perspective taking and PTSD symptoms; and independent self-construal mediated the relationships between fatalism and perspective taking and PTSD symptoms. However, cultural group did not moderate these conditional indirect effects. Cultural group did moderate the conditional indirect effects for soothing and social modelling. For the Malaysian group, interdependent self-construal mediated the relationships between soothing and social modelling with PTSD symptoms. However, these indirect effects were not observed for the Australian group.

Previous research suggests that members of Asian cultures value fatalism, specific cultural beliefs about adversity, suppression and interpersonal emotion regulation to a greater extent than those from Western cultural groups [7,11,12,16,22,23]. In line with this, the Malaysian group scored significantly higher on these appraisal types and emotion regulation strategies than the Australian group. This extends these past findings to a community sample of trauma survivors in Malaysia and Australia. We found the two groups did not differ significantly on trauma-specific appraisals (negative self, negative world, self-blame), which aligns with the findings of Bernardi and Jobson, who also found Asian Australians and European Australians did not differ significantly on these appraisals [21]. It is possible these trauma-specific appraisals are pan-culturally deemed relevant to trauma survivors.

Many of the appraisals and emotion regulation strategies were associated with PTSD symptoms. However, inconsistent with that predicted and previous research e.g., [21], cultural group did not moderate these associations, with the exception of self-blame. Therefore, in both cultural groups these appraisal and emotion regulation types were associated with PTSD symptoms. This is potentially an important finding. Research repeatedly demonstrates the processes that have little relevance for Asian trauma survivors [19,20,21,22]. This study provides initial evidence for the appraisal types and emotion regulation strategies that may have relevance for the psychological adjustment of Asian trauma survivors. Specifically, negative self, negative world beliefs, cultural beliefs about adversity, suppression and enhancing positive affect using interpersonal cues were associated with PTSD, regardless of cultural group. Regarding self-blame, the association was significantly stronger for the Malaysian group than the Australian group. This contrasts previous research that has found a stronger association between self-blame and PTSD for Western than Asian groups, e.g., [20,21]. It is not certain as to why this was the case. It may be that individuals from Western cultures tend to value presenting the self positively [49], while Asian cultures value identifying negative aspects of the self as this provides information about how to act accordingly to achieve interdependence [50]. Hence, for Malaysian trauma survivors, strong trauma-related self-blame may be more associated with PTSD as there may be additional burden associated with appraisals of letting down and/or failing others. Further research is needed in this area.

Our findings highlighted the importance of considering self-construal when considering cross-cultural PTSD research. While the Malaysian group reported significantly greater interdependence than the Australian group, they also reported greater independence than the Australian group, thereby demonstrating the need to also consider cultural variables. Some previous research has demonstrated that while Australia at a national level is more individualistic than Malaysia [51], when considering self-construal at the individual level Malaysians can score higher than Australians [52]. Additionally, the moderated mediation models showed that cultural group was only part of the story in understanding how cultural factors influence PTSD. We found there was an indirect pathway between certain appraisal types and emotion regulation strategies (cultural beliefs about adversity appraisals, reappraisals, enhanced positivity, perspective taking) and PTSD symptoms via interdependent self-construal.

Interpersonal emotion regulation considers how people utilize social cues to facilitate emotion regulation [18]. Most emotion regulation research in PTSD has focused exclusively on intrapersonal processes and much less attention has been given to examining interpersonal emotion regulation processes (processes relating to interdependence) [18]. Interdependent self-construal mediated the associations between interpersonal emotion regulation and PTSD symptoms. However, for soothing (i.e., seeking out others for comfort and sympathy) and social modelling (i.e., looking to others to see how they might cope with a given situation) these indirect effects were only observed for the Malaysian group. Thus, for Malaysian trauma survivors these emotion regulation strategies may be particularly associated with interdependence and PTSD. These findings highlight the need to extend PTSD emotion regulation research beyond intrapersonal processes to also consider interpersonal processes.

While fatalism and perspective taking were not significantly correlated with PTSD symptoms, there were indirect effects between these variables and PTSD symptoms through independent self-construal. However, these indirect effects were negative, indicating higher fatalism and perspective taking were associated with higher levels of independence which in turn were associated with lower levels of PTSD symptoms. We predicted these variables would be related to interdependence. However, perspective taking involves using others to be reminded not to worry and that others have it worse. Hence, this may be more related to independence as it involves using others as a social comparison [13]. Additionally, the fatalism questionnaire assessed an individual’s propensity to believe that one’s destiny is externally determined and thus, may be focused on the fatalism of the individual as a unique entity. Further research is needed exploring these dimensions in PTSD.

Theoretically, PTSD models emphasise the role of appraisals and emotion regulation in PTSD e.g., [2]. While there has been much empirical focus on Western appraisals and emotion regulation, there has been less focus on cultural beliefs about adversity and interpersonal emotion regulation. Thus, PTSD models need to consider cross-cultural research, explicitly noting that culture may influence the mechanisms underpinning PTSD and embed culturally different forms of appraisals and emotion regulation within theoretical conceptualisations [6]. Clinically, our findings indicate how PTSD treatments could be culturally tailored for those in Malaysia. Specifically, a focus on cultural beliefs about adversity, fatalism, and interpersonal emotion regulation may be of importance as clinical targets. Furthermore, for Malaysian client groups, there is a need to consider the association between appraisals/emotion regulation, self-construal values, and PTSD symptoms. Additionally, the self-construal findings indicate that these appraisals and emotion regulation may have relevance for Australian trauma survivors, especially those emphasising interdependence.

There are some limitations worth noting. First, the study was cross-sectional and thus, causality cannot be inferred. Second, the study included a community sample, and the generalizability of findings to a clinical sample still needs to be examined. Nevertheless, around 28% of each cultural group met provisional diagnosis for PTSD on the PCL-5, and average scores across cultural groups in anxiety and depression also exceed cut offs. Third, as noted above, the Malaysian group scored higher than the Australian group on independence, which is contrary to other research. Additionally, the Malaysian group tended to report higher levels on several constructs, and together these findings may reflect a measurement bias issue or an acquiescent bias. This needs to be considered when interpreting findings and suggests a need for further research. Fourth, while the study was adequately powered for the moderation analyses, a larger sample would benefit the moderated mediation analyses. Fifth, while we considered self-construal, it is important to recognise that Malaysian and Australian cultures differ in several other respects (e.g., religion, power hierarchy, holism/analytic thinking) that could influence findings and be considered in further research. Finally, while the groups did not differ in identified index traumas, trauma type (e.g., interpersonal vs. non-interpersonal, childhood) may have influenced findings more broadly.

## 5. Conclusions

Malaysian trauma survivors reported significantly greater fatalism, cultural beliefs about adversity, suppression and interpersonal emotion regulation than Australian trauma survivors. Regardless of cultural group, trauma-specific appraisals, cultural beliefs about adversity, enhancing positivity and suppression were associated with PTSD symptoms. Interdependent self-construal mediated the relationships between cultural adversity beliefs, enhanced positivity, perspective taking, soothing (Malaysian group only), social modelling (Malaysian group only) and PTSD symptoms. Independent self-construal mediated the relationships between fatalism, perspective taking and PTSD symptoms. These findings demonstrate the importance of considering self-construal and culture in understanding factors associated with PTSD.

## Figures and Tables

**Figure 1 ijerph-19-01163-f001:**
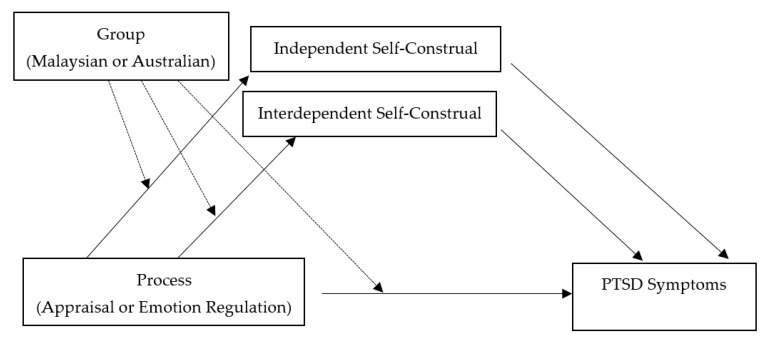
Depiction of the moderated mediation analyses.

**Figure 2 ijerph-19-01163-f002:**
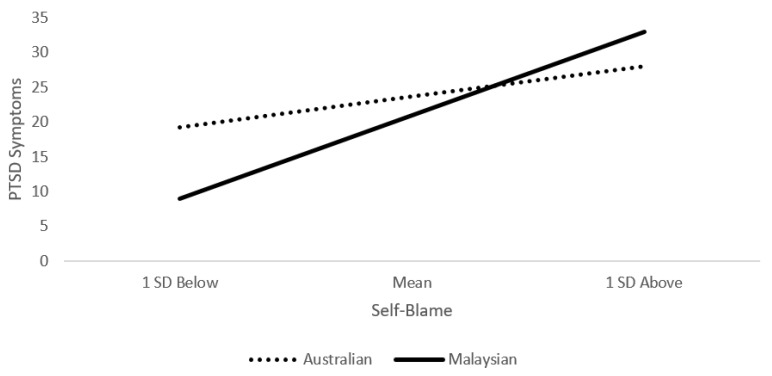
Simple slopes for the self-blame moderation analysis.

**Table 1 ijerph-19-01163-t001:** Group characteristics.

Variable	Australian	Malaysian	Group Comparison
Age (years)	31.66 (12.76)	25.48 (6.78)	*t* (216) = 4.51 **
Gender ^a^	19:87:1	28:92:1	*χ2* (2, 228) = 1.01
Education ^b^	26:21:38:20:2	8:10:78:19:6	*χ2* (4, 228) = 28.50 **
Religion ^c^	36:56:0:6:0:7	2:19:33:38:16:6	*χ2* (5, 219) = 120.86 **
Trauma type (*n*)			*χ2* (5, 228) = 4.65
Accident/Sudden death	48	59	
Non-Sexual Assault	18	14	
Sexual Assault	17	13	
Life-Threatening Illness	16	20	
War/Conflict/Kidnapping	3	3	
Natural Disaster	5	12	
Time since trauma (years)	9.90 (11.31)	6.13 (5.49)	*t* (226) = 3.26 **
PTSD Symptoms	22.96 (17.69)	22.48 (18.39)	*t* (226) = 0.20
Depression Symptoms	13.94 (4.20)	12.62 (5.02)	*t* (226) = 2.14
Anxiety Symptoms	11.56 (5.73)	11.70 (5.62)	*t* (226) = 0.29
Independence	4.57 (0.77)	5.08 (0.73)	*t* (226) = 5.04 **
Interdependence	4.56 (0.73)	5.09 (0.60)	*t* (226) = 6.01 **

Note: ^a^ Male:Female:Non-Binary. ^b^ Secondary:Post-Secondary:Undergraduate:Postgraduate:Other/Prefer not to say. ^c^ None: Christian:Buddhism/Taoism: Islam:Hinduism:Other. ** *p* < 0.001.

**Table 2 ijerph-19-01163-t002:** Group differences on appraisal types and emotion regulation strategies.

Variable	Australian	Malaysian	Group Comparison
**Appraisal Types**			
Self-Blame	12.58 (8.08)	14.41 (6.63)	*F*(1, 179) = 0.04, *η_p_*^2^ = 0.001
Negative Self	51.05 (25.36)	53.15 (25.62)	*F*(1, 179) = 0.86, *η_p_*^2^ = 0.005
Negative World	24.96 (11.08)	27.95 (10.07)	*F*(1, 179) = 0.06, *η_p_*^2^ = 0.001
Fatalism	18.46 (4.15)	21.28 (4.66)	*F*(1, 179) = 7.99 **, *η_p_*^2^ = 0.04
Cultural beliefs about Adversity	34.47 (5.37)	37.99 (5.57)	*F*(1, 179) = 14.85 **, *η_p_*^2^ = 0.08
**Emotion Regulation Strategies**			
Reappraisal	26.24 (6.45)	30.95 (5.38)	*F*(1, 206) = 20.25 **, *η_p_*^2^ = 0.09
Suppression	15.27 (5.06)	17.88 (4.69)	*F*(1, 206) = 5.05 *, *η_p_*^2^ = 0.02
Enhance positivity	18.79 (4.05)	17.87 (4.96)	*F*(1, 206) = 1.26, *η_p_*^2^ = 0.006
Perspective taking	11.11 (5.09)	14.62 (4.89)	*F*(1, 206) = 19.44 **, *η_p_*^2^ = 0.09
Soothing	13.46 (5.16)	14.78 (5.34)	*F*(1, 206) = 1.55, *η_p_*^2^ = 0.01
Social modelling	14.25 (4.80)	16.62 (4.81)	*F*(1, 206) = 5.06 *, *η_p_*^2^ = 0.02
IER	57.60 (13.98)	63.90 (17.29)	*F*(1, 206) = 4.72 *, *η_p_*^2^ = 0.02

* *p* < 0.05, ** *p* < 0.01. Note: PTCI = Post-traumatic cognitions inventory. IER = Interpersonal emotion regulation total.

## Data Availability

Data are available by contacting the authors.

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
