# Peer review of "Examining Cultural Differences in the Associations between Appraisals and Emotion Regulation and PostTraumatic Stress Disorder in Malaysian and Australian Trauma Survivors"

_ijerph, 2022, doi:10.3390/ijerph19031163_

Round 1
Reviewer 1 Report
In general, I have to give congrats to the authors for an excellent article. The theme is relevant, and the discussion and the conclusions are very well established and supported.
In terms of suggestions, there are a few:
In the abstract:
- please specify better the difference between the two populations - this mentioned in the Introduction;
- it´s useful to discriminate what survey it was used (you can use a brief summary of point 2.3.1.);
In the Introduction:
- line 1: PTSD, as codified in DSM/ICD, it´s a psychiatric condition;
- line 137 is unnecessary;
- very good framework constructed for hypothesis;
in the data analysis:
- congrats on the excellent details;
- it could be objectionable the content of note 1 - the exclusion of important factors as age or religion in this data analysis;
In the Results:
- the presentation of statistical data in the text it´s a bit confusiing - please reformulate;
- very interesting results for hypothesis 2.
Author Response
Thank you for your time in reviewing our paper and helpful feedback. We have responded to each point as detailed below and highlighted in the manuscript.
1) We have added to the abstract further details about the groups: "This study examined cultural differences in the associations between appraisals, emotion regulation and PTSD symptoms using trauma survivors from an individualistic Western culture (Australia) and a collectivistic Asian culture (Malaysia). Trauma survivors (N=228; 107 Australian with European cultural heritage, 121 Malaysian with Malay, Indian or Chinese cultural heritage)... "
2) Abstract: We have added: "(PTSD Checklist for the DSM-5 with Life Events Checklist)"
3) Introduction: We have changed 'psychological' to 'psychiatric'
4) Introduction: We removed the line before 'Current Study'
5) We have clarified Note 1: "However, it is worth noting that when these variables were included as covariates a similar pattern of results emerged."
6) We have clarified and re-worked the results section so it is simpler and easier to follow.
Reviewer 2 Report
Review of article:
Examining Cultural Differences in the Associations between Appraisals and Emotion Regulation and Posttraumatic Stress Disorder in Malaysian and Australian Trauma Survivors
Strengths of the article:
- The topic is interesting and important, and the article makes an important contribution to the research literature that discusses mental health in the cultural context. To date, very few articles discussed the influence of culture on processes of appraisals and emotional regulation that play a central role in posttraumatic stress disorder (PTSD), despite research demonstrating cultural differences in everyday appraisals and in posttraumatic stress disorder (PTSD).
- The literature review on posttraumatic stress disorder (PTSD) is thorough, comprehensive and up-to-date.
- Method – I was very impressed with the detailed research procedure, that was carried out according to accepted scientific rules: approval of the ethics committee; the manner in which participants were recruited; definition of the criteria for testing the participants; the variables for testing the symptoms of posttraumatic stress disorder, and more.
The main weakness of the article, that necessitates correction
- The cultural aspect comprises the core of the article’s aim. However, the discussion of this issue is deficient. The author does indeed present, within the framework of the Introduction, the two ethnic groups discussed in the article: the Malaysian and the Australian. However, the discussion on their cultures is only presented in a research context and from the viewpoint of posttraumatic stress disorder. This discussion is missing a broad sociological-anthropological background about these cultures, which is not related to the context of PTSD. For example, beliefs, rituals, language, traditions, unique cultural values, family and community aspects. The absence of this essential information may lead the researcher to a narrow reference in analyzing the findings and reaching conclusions, that refers solely to the cultural characteristics that were presented in the context of PTSD, and does not take additional cultural characteristics into account.
The author is therefore asked to present a comprehensive cultural background for the two ethnic groups, that is disengaged from the discussion on posttraumatic stress disorder. Such a chapter may contribute to understanding and analysis of the phenomenon of posttraumatic stress disorders in the cultural context of each group; to understanding the difference between the groups; and to affording a foundation for the research conclusions that point to the cultural context of psychological distresses.
Author Response
Thank you for your time in reviewing our manuscript. This is a very valid and important point.
We have added to the Introduction
Malaysia is a Southeast Asian country. After more than a century of British colonial rule, Malaysia gained independence in 1957 [27]. Contemporary Malaysian society is influenced by Malay, Chinese and Indian values and traditions [28]. Around half of the population of Malaysia is Malay, with minorities of Indian, Chinese and indigenous peoples [29]. Malaysia’s official language is Malay and official religion is Islam [28, 29]. Malaysia tends to be accepting of hierarchical order in society and values social harmony and the honouring of cultural norms and traditions [26]. Of relevance to the current study, Malaysians tend to hold a fatalistic worldview [28,29], Malaysian culture has been significantly influenced by Chinese beliefs and values [30, 31], and Malaysia is a collectivistic, interdependent society [26].
Australia was colonized by the British in the late 18th century, resulting in Aboriginal and Torres Strait Islander peoples being dispossessed of their lands and subject to genocidal practices and policies [32]. The social composition of the country was dramatically altered and a Western, European culture became dominant [33].The first half of the 20th century focused on facilitating ‘white’ migration [34]. However, in modern Australia migrants come from Asia, the Americas and Africa, with approximately 26% of Australian residents being born overseas [33]. The official language of Australia is English and Australia is considered a secular country, with Christianity being the dominant religion introduced by the British colonial settlers [33]. Australian society values emphasise egalitarianism [26]. Australia is a highly individualistic society [26].
Round 2
Reviewer 2 Report
I confirm this revised version